https://doi.org/10.1038/s42003-019-0660-7　　**OPEN**

# Properties and efficient scrap-and-build repairing of mechanically sheared 3′ DNA ends

Yoshiyuki Ohtsubo [1]*, Keiichiro Sakai[1], Yuji Nagata[1] & Masataka Tsuda[1]

Repairing of DNA termini is a crucial step in a variety of DNA handling techniques. In this study, we investigated mechanically-sheared DNA 3′-ends (MSD3Es) to establish an efficient repair method. As opposed to the canonical view of DNA terminus generated by sonication, we showed that approximately 47% and 20% of MSD3Es carried a phosphate group and a hydroxyl group, respectively. The others had unidentified abnormal terminal structures. Notably, a fraction of the abnormal 3′ termini (about 20% of the total) was not repaired after the removal of 3′ phosphates and T4 DNA polymerase (T4DP) treatment. To overcome this limitation, we devised a reaction, in which the 3′− > 5′ exonuclease activity of exonuclease III (3′− > 5′ exonuclease, insensitive to the 3′ phosphate group) was counterbalanced by the 5′− > 3′ polymerase activity of T4DP. This combined reaction, termed "SB-repairing" (for scrap-and-build repairing), will serve as a useful tool for the efficient repair of MSD3Es.

[1] Department of Molecular and Chemical Life Sciences, Graduate School of Life Sciences, Tohoku University, 2-1-1 Katahira, Sendai 980-8577, Japan.
*email: yohtsubo@ige.tohoku.ac.jp

The repair or polishing of DNA termini is a crucial step in the handling of DNA fragments, especially those generated by mechanical shearing, such as sonication, to efficiently conduct subsequent enzymatic reactions. DNA sonication[1] has been widely used for random subcloning of DNA for shotgun DNA sequence analysis[2], the preparation of next-generation sequencing (NGS) libraries for genomic and meta-genomic sequencing[3–5], Tn-seq analysis[6], microarray analysis[7], and ChIP analysis for mapping binding sites of DNA binding proteins[8].

The DNA terminal structures generated by sonication and other strand break mechanisms have been previously investigated[9–11]. Previous studies have suggested that most of the sonication-mediated degradation occurs by the rupturing the C–O or P–O linkage within the sugar-phosphate backbone, resulting in 3' or 5' phosphate groups, or other possible structures (e.g., 2',3'-double bonds)[9]. The review article by Elsner and Lidblad has described, "Breaks in the DNA helix occur mainly between oxygen and carbon atoms, resulting in DNA fragments with a phosphorylated 5' end and a free alcohol at the 3' end".[10]. This description is a widely accepted theory of sonication-generated DNA termini. However, this description is based on a misinterpretation of an article by Richardson[12], which merely showed that of 5'-OH and 5' phosphate termini generated by sonication, 95% were 5' phosphate termini. In the article, other kinds of 5' terminal besides -OH and phosphate were not taken into consideration, and no experiments were conducted to investigate the 3' terminus. As is exemplified in this study, our experimental results are not consistent with this description by Elsner and Lindblad, suggesting that the features of mechanically-sheared DNA termini have long been misinterpreted.

In our previous report, we showed that reverse transcriptase from Molony Mouse Leukemia Virus[13] (MMLV-RT) had an extraordinarily high tailing activity to append, in a template-independent manner, several nucleoside monophosphates to blunt DNA 3' termini generated by enzymatic digestion[14]. The tailing activity was observed for substrates with or without a 5' phosphate group on the opposite strand. The activity was further enhanced in the presence of specific enhancing chemicals[15], and the generated 3' tail consisting of GMPs (guanosine monophosphates) was found to serve as a good substrate for the CIS (clamping-mediated incorporation of single-stranded DNA with concomitant DNA syntheses) reaction. In the CIS reaction, a single-stranded DNA carrying several cytidine monophosphates (CMPs) at its 3' terminus was incorporated by concomitant complementary strand synthesis, which began at the 3' terminus of the G-tail[16]. The efficiency of the CIS reaction, using a DNA terminus generated by PvuII digestion, was as high as 98%. Our attempt to apply G-tailing and a subsequent CIS reaction to sonicated genomic DNA after a T4 DNA polymerase (T4DP) blunting reaction, however, yielded poor results (c.a. 30% efficiency), suggesting that establishing a better method for repairing the sonicated DNA 3' terminus is key to a successful CIS reaction.

Aigrain et al. reported that NGS-sequence library preparation efficiencies using sonication-generated DNA fragments were as low as 3%–20%, which is in marked contrast to the approximate 100% efficiency of enzymatically digested-DNA fragments[17]. This further highlights the need for establishing an adequate method for the repair of MSDEs.

In this study, to gain insight into the properties of sonication-generated DNA termini and to develop a method to efficiently and thoroughly repair these termini, we prepared a mixture of model DNA substrates with mechanically-sheared DNA ends. We present evidence that 3' phosphates, as well as unknown 3'-end-structures resistant to T4DP treatment in the presence of dNTP, both represent obstacles to DNA end-repairing. The latter terminus represents 20% of the generated MSDEs. Finally, we also present a method for the removal of 3' phosphates and T4DP-resistant structures.

## Results

**Preparation of DNA substrates with mechanically sheared ends**. To efficiently analyze MSDEs, we prepared a model DNA substrate mixture. In brief, a 560-bp DNA fragment carrying a FAM label at its 5' terminus was PCR-amplified, purified, and fragmented by sonication. The resulting fragments were size-fractionated by polyacrylamide gel electrophoresis, and different size fractions were excised and used. In this study, we show data obtained from a fraction, denoted as H4, which contained DNA fragment sizes ranging from 100 to 150 bp. Although the H4 fraction was one of the most extensively studied fractions, consistent data were obtained by using different fractions.

**Detection of FAM-labeled strand**. DNA samples were mixed with HiDiLIZ500, a mixture of DNA size standards labeled with LIZ fluorophore and deionized formamide, and analyzed through capillary sequencer under denaturing conditions. Figure 1a shows an electropherogram of the H4 mixture. The electropherogram exhibited a disordered pattern, as seen by the uneven peak spacing and shallow bottoms observed between the peaks. The disordered pattern suggested that DNAs with different chemical structures are contained in the mixture. Figure 1b shows a merged view of ten electropherograms obtained by repeated analysis ($n = 10$) of the H4 mixture. Note that the overall peak patterns are reproducible, showing that the features observed in the electropherogram do not represent experimental noise.

**Enzymatic reactions on MSDEs**. Panels c–g in Fig. 1 show electropherograms obtained after different enzymatic reactions of the H4 mixture, including the newly developed scrap-and-build-repairing (SB-repairing) method (see below). T4DP treatment (Fig. 1c) for 5 min at 37 °C resulted in an altered, but still disordered pattern, indicating that the T4DP was unable to react with MSDEs efficiently, resulting in the most of the mixture being unchanged. Equivalent data were obtained by extending the reaction time to 30 min, conducting the reaction at 20, 30, or 42 °C for 30 min, or conducting the reaction in the presence of lower dNTP concentrations (25 or 2.5 μM each of dNTPs) (Supplementary Fig. 1). SAP treatment for 30 min at 37 °C resulted in altered but still disorderly spaced peaks (Fig. 1d). SAP changed the 3' phosphate terminus to a 3'-OH terminus, and the remaining disorderly spaced peaks indicated that the DNAs after SAP reaction had at least two types of 3' structure. One type is the 3'-OH terminus, and other possible, yet unidentified structures might include those derived from the breaking of the C–C bond between the 4' and 5' carbon atoms, abasic structure, and 2', 3'-double bonds. Hereafter, the unidentified structures will be collectively referred to as "abnormal structures". Following SAP treatment, the DNA mixture was treated with T4DP (SAP-T4DP; Fig. 1e), which resulted in more orderly spaced peaks, suggesting that the 3' termini changed towards a normal (3'-OH) state (compare Fig. 1c, e). As shown below, however, there still remains a fraction of DNA, which does not serve as substrates of MMLV-RT and terminal deoxynucleotidyl transferase (TdT). Those DNAs might carry the 3' abnormal structure in blunt or recessed configurations, to which T4DP does not exert its exonuclease activity in the presence of dNTPs.

**Use of exonuclease III for blunting**. Although SAP treatment followed by T4DP reaction changed the MSDEs towards a normal state, the abnormal terminal structures were supposed to be still present in blunt or 3'-recessed configuration. To remove the

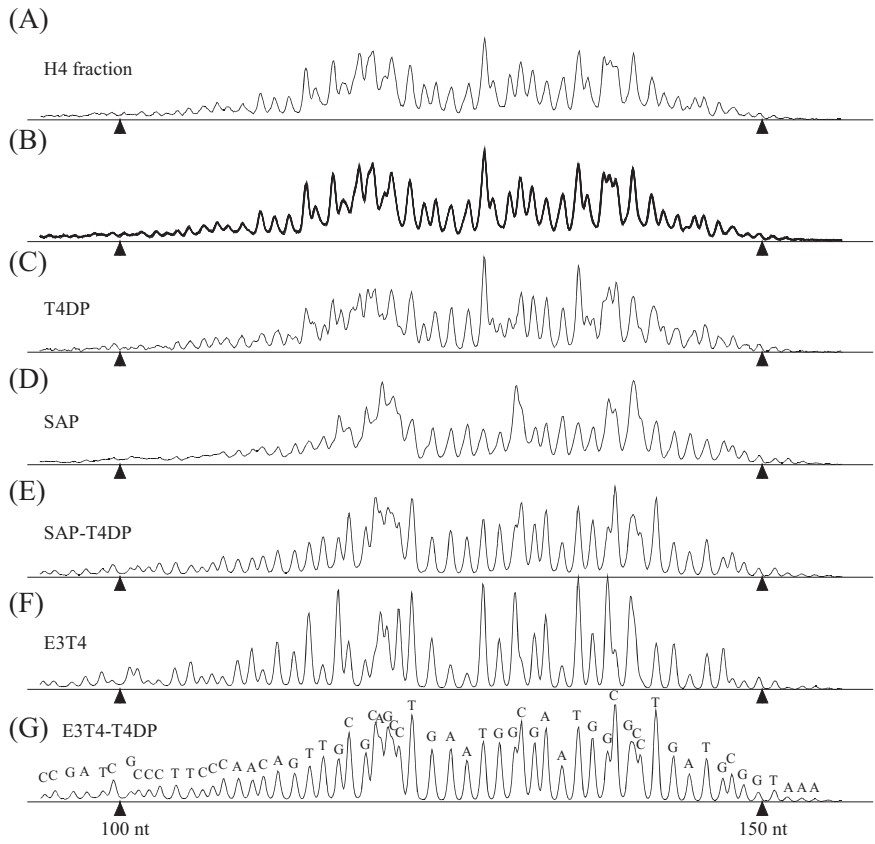

**Fig. 1** Analysis of MEDEs using a capillary sequencer. FAM-labeled DNA samples were analyzed using a capillary sequencer after mixing with LIZ500 size standard-supplemented HiDi-formamide (HiDi-LIZ500). Two peaks of 100 and 150 nucleotides from the LIZ size standard, used to calibrate the electropherogram, are indicated by filled triangles. **a** H4 fraction DNA mixture before reaction. **b** Merged view of ten results obtained by analyzing H4 fraction DNA mixture independently. **c–g** Data obtained after different enzymatic treatments indicated on the left. Samples were purified using a DNA clean-up column before mixing with HiDi-LIZ500. T4DP, T4 DNA polymerase; SAP shrimp alkaline phosphatase; E3T4, combined treatments with exonuclease III and T4DP; SAP-T4DP, SAP-treated sample purified and treated with T4DP; E3T4-T4DP, E3T4-treated sample purified and treated with T4DP. In panel **g**, each peak is labeled with a corresponding base. All data were y-axis scaled so that sums of FAM peak areas are apparently even across the panels

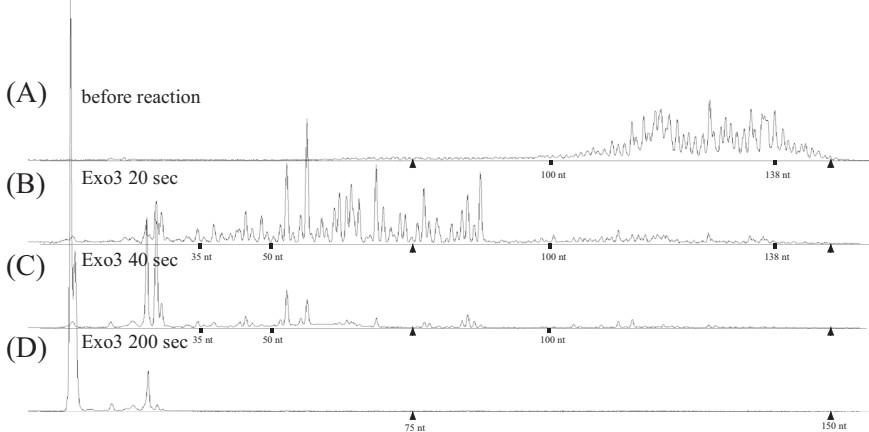

**Fig. 2** Exonuclease III treatment of MEDEs resulting in the complete removal of MSDEs. Exonuclease III was added to an H4 DNA mixture (10 nM) at a final concentration of 4 unit/μL. **b–d** At each indicated time point, a fraction (0.5 μL) of the reaction mixture was sampled to HiDi-LIZ500 and analyzed. Two calibrator peaks from the LIZ size standard (75 and 150 nucleotides) are indicated by filled triangles. The relevant size marker locations are indicated. Panel **a** shows identical data to that shown in Fig. 1a

abnormal terminal structures, exonuclease III (non-processive 3′−>5′ exonuclease) was used. For the first step, its ability to degrade DNA with MSDEs was confirmed. The exonuclease III treatment of the H4 mixture resulted in the complete degradation of the FAM-labeled strand (Fig. 2), demonstrating the sensitivity of all MSDEs to exonuclease III. Notably, no SAP treatment was required. In the second step, to prevent complete degradation by exonuclease III, we applied a mixture of exonuclease III and

T4DP. We expected that 3' to 5' exonuclease activity of exonuclease III is counter-balanced by DNA polymerization activity of T4DP, keeping the overall DNA lengths unchanged, and that these two opposite activities would polish MSDEs. We tested different ratios of exonuclease III to T4DP and found that the combination of 1 unit of exonuclease III and 1.25 unit of T4DP activity maintained the balance.

Hereafter, we refer to this treatment as "E3T4 treatment." Fig. 1f shows the H4 mixture after E3T4 treatment. Since the E3T4 treatment resulted in a fraction of DNA with a recessed terminus, the H4 mixture after E3T4 treatment was purified and further reacted with T4DP (E3T4-T4DP; Fig. 1g).

As seen in the merged view (Supplementary Fig. 2), E3T4-T4DP treatment resulted in much sharper peaks than SAP-T4DP treatment, indicating that E3T4-T4DP treatment polished DNA termini more thoroughly than SAP-T4DP treatment.

**Evaluation of repaired DNA termini by CIS reaction**. We investigated the reactivity of the 3' termini by G-tailing and a subsequent CIS reaction[16]. In a series of two reactions, the FAM-labeled strand was first extended by G-tailing and further extended by DNA polymerization along a single-stranded DNA (Fig. 3a). The electropherogram of the initial substrate is shown in

Fig. 3b. When the G-tailing and subsequent CIS reaction product of the E3T4-T4DP pretreated sample (Fig. 3c) was compared to the SAP-T4DP pretreated sample (Fig. 3d), the latter exhibited more signals representing unreacted DNAs. A sample pretreated with T4DP (Fig. 3e) showed an overall reaction efficiency value of $33 \pm 0.6\%$ (values are average and standard deviation of independent reactions; $n = 3$). The DNA length was changed by 52 nucleotides, which was sufficient to differentiate the reaction product from the initial substrate for Fig. 3e. However, quantitative analysis of unreacted peaks shown in Fig. 3c, d were difficult due to the presence of smaller DNAs in the H4 fraction (see the signals around 75 nt) whose products severely interfered with the unreacted peaks.

**Evaluation of repaired DNA terminus by TdT**. To complement the quantitative limitation, we applied terminal deoxynucleotidyl transferase (TdT), which appends dozens to hundreds of nucleotides to 3'-OH terminus of DNA, irrespective of whether the 3' is recessed, protruding, or blunt-ended[18]. TdT treatment lengthens DNA fragments with normal 3'-OH terminus while maintaining the length of DNA with abnormal termini. We conducted different treatments (SAP, SAP-T4DP, and E3T4-T4DP) followed by TdT treatment. Figure 4 demonstrates that,

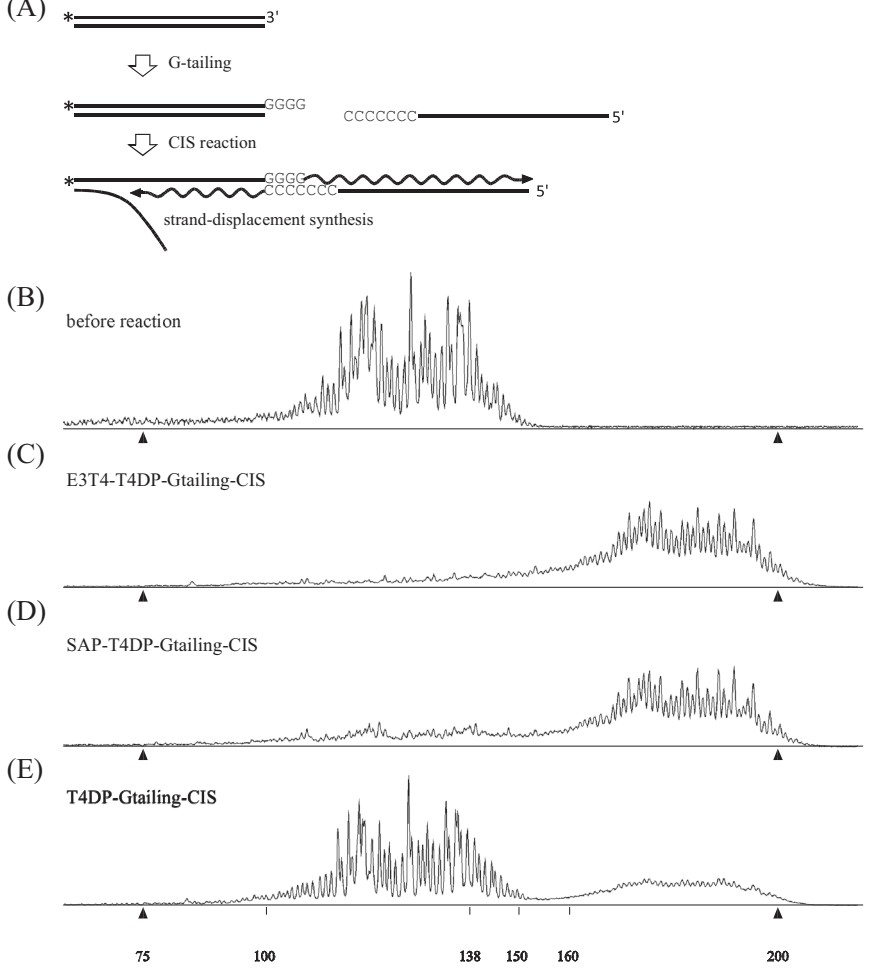

**Fig. 3** G-tailing and CIS treatments. **a** Schematic drawing of G-tailing and CIS reactions. G-tailing reaction using MMLV-RT conducted in the presence of 4 mM MnCl2 and 200 mM deoxycytidine appended approximately four deoxyguanosine monophosphates to the 3'-OH terminus. For G-tailing, a single-stranded DNA with serial deoxycytidine monophosphates was incorporated with concomitant DNA polymerization mediated by MMLV-RT. Asterisk (*) indicates 5' FAM fluorophore. **b** H4 fraction DNA before reaction (identical to that shown in Fig. 1a). **c–e** 50 fmol of H4 DNA fraction was subjected to serial reactions as indicated. Between the two enzymatic treatments, and before mixing with HiDi-LIZ500, the samples were purified using the DNA clean-up column

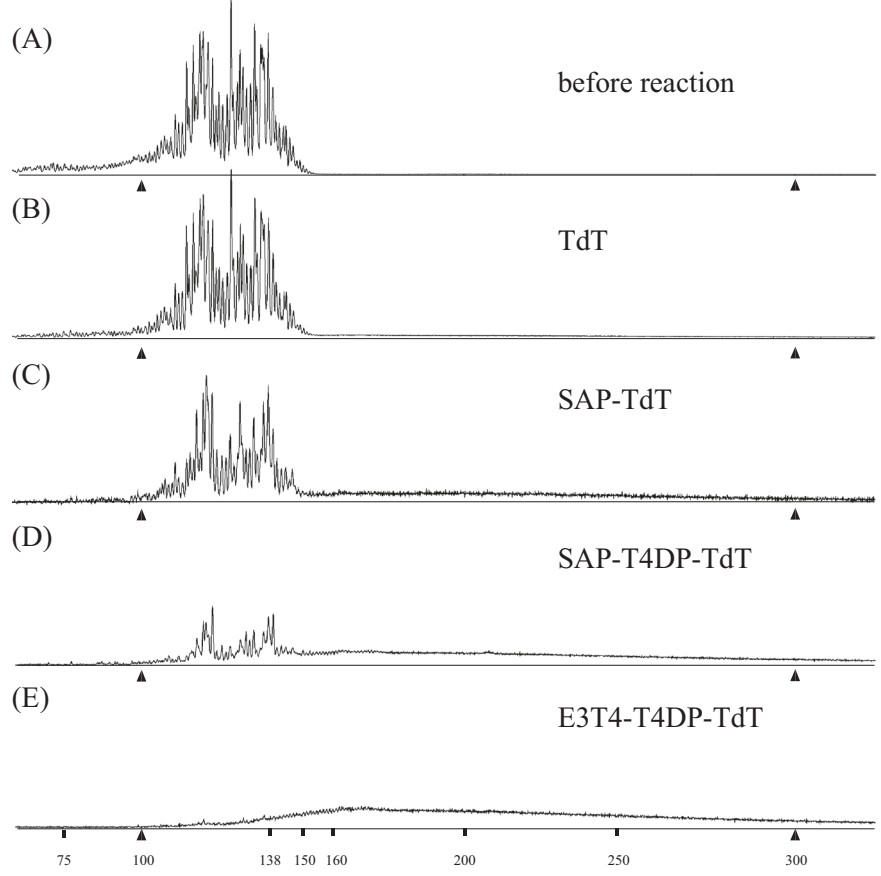

**Fig. 4** Terminal deoxynucleotidyl transferase (TdT) treatment of samples pretreated with different enzymes. H4 DNA mixture containing 25 fmol of FAM-labeled DNA was subjected to different enzymatic treatments, followed by TdT treatment. The panels show the samples before treatment (**a**, identical to that shown in Fig. 1a); samples treated with TdT alone (**b**); samples treated with SAP followed by TdT (**c**); samples treated serially with SAP, T4DP, and TdT (**d**); samples treated serially with a combination of exonuclease III and T4DP, T4DP alone, and TdT (**e**). Between the two enzymatic treatments, the samples were purified using the DNA clean-up column. Following the TdT treatment, the samples were purified using a DNA clean-up column before mixing with HiDi-LIZ500. All data were y-axis scaled so that sums of FAM peak areas are apparently even across the panels. Two calibrator peaks from the LIZ size standard (100 and 300 nucleotides) are indicated by filled triangles, and relevant size marker locations are also indicated

while the E3T4-T4DP-TdT-treated sample (Fig. 4e) showed a marked decrease in the peaks representing the initial substrates, the SAP-TdT-treated and SAP-T4DP-TdT-treated samples exhibited wider peak areas of the initial substrates. If we compare Fig. 4c, d, the residual peaks are diminished by T4DP treatment in Fig. 4d, indicating the removal of abnormal ends and formation of 3′ -OH ends by T4DP. The remaining peaks in Fig. 4d represent abnormal ends recalcitrant to T4DP, possibly placed in blunt or recessed configurations.

The quantification of the ratio of substrate not extended by TdT to total substrate was not straightforward due to the overlap between the substrate and product peaks. To calculate the residual substrate ratio, we assumed that the signals observed in the E3T4-T4DP-TdT-treated sample were all extended products (see below), and the product peak profiles were the same for the other samples. As a result, we obtained residual substrate ratios of 35.0 ± 4.0% (SAP-TdT), 18.9 ± 3.2% (SAP-T4DP-TdT), and 79.9 ± 2.1% (TdT alone) (values are averages and standard deviations of independent reactions; $n = 3$).

**Quantification of 3′ phosphate ends**. To quantify the fraction of MSDEs carrying 3′ phosphate group, the H4 DNA was reacted with T4DP in the absence of dNTPs, a condition that promotes 3′− > 5′ exonuclease activity of T4DP. T4DP has been shown to be incapable of degrading single-strand DNA with 3′ phosphate,

even in the absence of dNTPs[19]. Figure 5a shows DNA fragments immediately after the addition of T4DP. As shown in Fig. 5b, the treatment for 5 min resulted in two major peaks representing the degradation products, possibly one and two nucleotides with FAM fluorophore, and orderly spaced T4DP-recalcitrant peaks. In a parallel experiment using SAP-treated H4 fraction led to complete degradation (Fig. 5e), suggesting that the T4DP-recalcitrant peaks represented DNA molecules with 3′ phosphate groups, and showing that abnormal 3′ termini are sensitive to the exonuclease activity of T4DP in the absence of dNTPs.

The quantification of the ratio of T4DP-recalcitrant DNAs to total DNAs was conducted. We observed that upon the addition of T4DP in the absence of dNTPs, approximately half of the substrate DNA was degraded rapidly, giving rise to two major degradation product peaks. The remaining peaks were orderly spaced, indicating that they represent DNA molecules sharing the same 3′ structure. Although the reason is not clear, several peaks among the orderly spaced peaks were found to diminish after a prolonged incubation (Fig. 5c). Thus, we measured the residual peak area at a time point when SAP-pretreated DNA was completely degraded by T4DP (Fig. 5e). Notably, the total FAM signal increased as the two major degradation product peaks emerged, possibly due to the nature of the smaller degradation product, which can be easily electro-injected into a capillary. Therefore, we added another FAM-labeled double-strand DNA to the HiDiLIZ500 and used its peak area to calculate the proportion

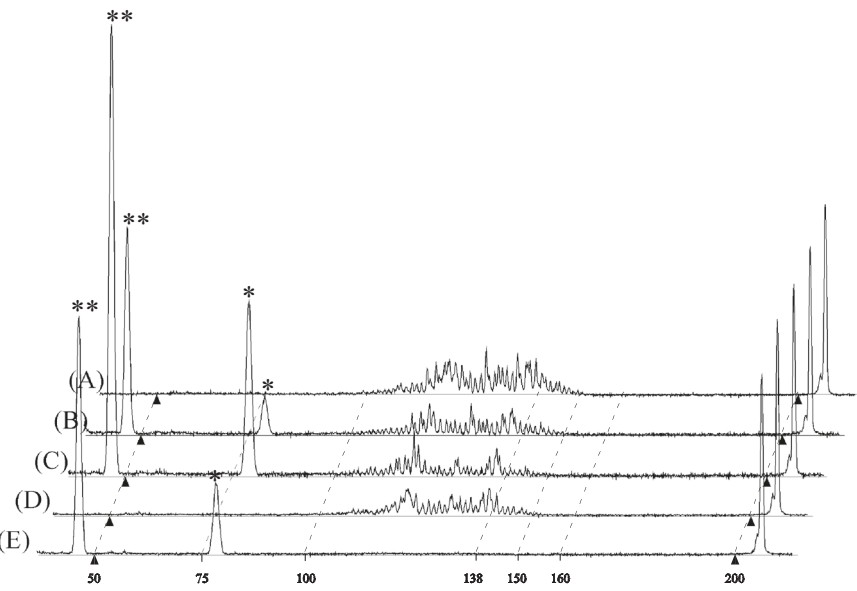

**Fig. 5** T4DP treatment in the absence of dNTPs. 10 μL of H4 DNA mixture containing 50 fmol of FAM-labeled DNA was incubated without (**a–c**) or with (**d**, **e**) SAP for 30 min at 37 °C. Then, 0.5 units of T4DP were added. The reactions were conducted in the absence of dNTPs. After the indicated time intervals, a portion of the reaction mixture was sampled to HiDi-LIZ500 supplemented with 201 bp FAM-labeled DNAs. Data from samples taken immediately after (**a** and **d**), 5 min (**b** and **e**), and 20 min (**c**) after T4DP addition are shown. All data were y-axis scaled so that FAM peak areas of the 201 bp DNA (right most peak) are apparently even across the panels. Two calibrator peaks from the LIZ size standard (50 and 200 nucleotides) are indicated by filled triangles. The relevant size marker locations are also indicated. Peaks observed at a marker size of about 50 nt and 75 nt represent the degradation product of the FAM-labeled strand, which may be two nucleotides plus FAM (5'-FAM-AA-3', indicated by asterisks (\*\*)) and one nucleotide plus FAM (5'-FAM-A-3', indicated by astersisk (\*)), respectively. Electric charge of the FAM moiety might have pronounced effects on the mobility of DNA fragment as the nucleotide length decreases. The slight height increase of some peaks in (**c**) might be due to injection profile change due to prolonged incubation

of T4DP-recalcitrant molecules. The analysis of the H4 fraction indicated that $46.7 \pm 3.6\%$ of MSDEs contained 3' phosphate (values are average and standard deviation of independent reactions; $n = 3$).

## Discussion

Our analysis showed that DNA 3' termini generated by sonication could be roughly classified into three categories: terminus with 3'-OH group, terminus with a 3' phosphate group, and terminus with unidentified abnormal structure. We observed DNA molecules with 3' phosphate as persistent peaks that remained after T4DP treatment conducted in the absence of dNTPs (Fig. 5b). We also observed DNA with a 3' abnormal terminus (Fig. 4c). There appears to be at least two types of abnormal structures, since the remaining peaks after the SAP-TdT treatment of the sonicated DNA were not orderly spaced. Because the distribution of the remaining peaks in Fig. 4c and Fig. 5b seems not to be biased, the formation of abnormal and 3' phosphate termini might not be DNA sequence specific. Formation of those 3' end types were confirmed by sonication at a milder set of conditions that was recommended by the manufacturer to fragment genomic DNAs into 1500 bp in average (Supplementary Fig. 3), suggesting that such ends are generally generated at conventional sonication intensities.

The overall proportions of each type of 3' structure in our sample are shown in Fig. 6. In our sample, 35% of the termini had abnormal termini, and importantly, more than half (18.9% of total) were not repaired by T4DP treatment in the presence of dNTPs. In addition, more than 45% of 3' termini carried a phosphate group as calculated by peak areas remaining after T4DP treatment without dNTPs. The TdT treatment of the H4 fraction showed that only 20% of termini were accessible by TdT, i.e., termini with a 3'-OH group (Fig. 4b). These findings are

inconsistent with Elsner and Lindblad's statement that sonication results in DNA fragments with a phosphorylated 5'-end and a free alcohol at the 3'-end[10], which may have resulted from the misinterpretation of the report by Richardson[12].

These ratios may vary depending on the nucleotide composition, shearing devices, and buffer conditions, and etc. Moreover, although we have focused on the sonication-generated DNA termini in this study, genomic DNAs are sheared during the preparation procedures. Furthermore, several other means, besides sonication, are used to shear DNA mechanically, and the ratios for these procedures are unknown. In this regard, for DNA termini end-repair, the method described here for the efficient repair of 3' termini could be an ideal approach (see below).

Further studies are needed for the identification of the chemical structure of the abnormal ends; however, a beta elimination product containing a 2',3'-double bond is a possible candidate. Strand breakage at abasic sites are known to result in a 2',3'-double bond[20], and free DNA bases have been detected upon sonication[11]. Two types of 3'-terminal chemical structures generated by ionizing radiation have been reported: 3' phosphate and 3' phosphoglycolate[19]. The 3' phosphoglycolate terminus did not change into an -OH terminus upon phosphatase treatment[19]. In this study, we observed that all termini became sensitive to T4DP upon SAP treatment (Fig. 5e), indicating that the abnormal structures observed in this study do not include 3' phosphoglycolate.

DNA polymerases require a 3'-hydroxyl group as a primer for DNA synthesis, and 3' phosphate does not serve as primer for DNA synthesis. It also does not serve as a substrate for ligases. Therefore, 3' phosphate must be removed to ensure the efficiency of the downstream reactions. However, the need for 3' phosphate removal appears to be ignored in many studies or commercial kits for DNA end-repairing, which may be a consequence of the

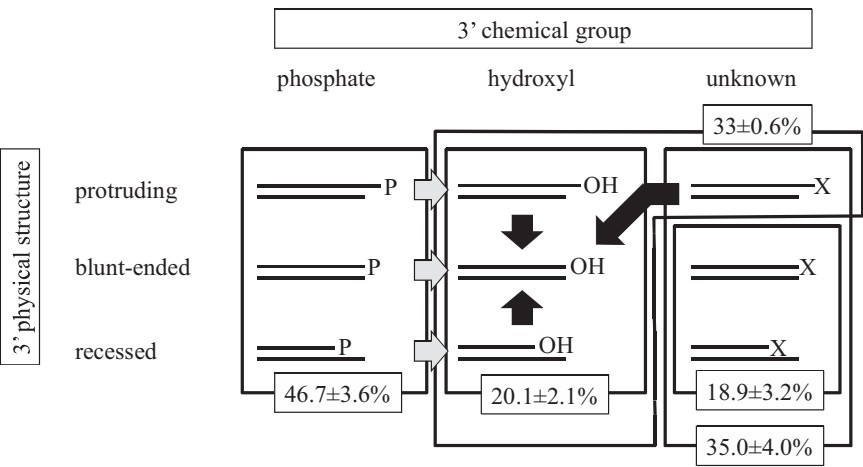

**Fig. 6** Proportions of terminal chemical structures. DNA termini were classified into nine groups based on two criteria. The 3' DNA terminus can be protruding, blunt-ended, or recessed with respect to the 5'-end of the opposing strand. The 3' DNA terminus can carry a phosphate group, hydroxyl group, or an as of yet unidentified abnormal chemical structure represented by "X" here. The gray and black arrows indicate the reactions mediated by SAP and T4DP, respectively. The indicated values (average and standard deviation from triplicate independent reactions) were obtained by reacting H4 fraction DNAs with different enzymes. See the text for further details

prevalent view of DNA termini generated by sonication[10]. The 3' phosphate may have been hidden due to concomitant usage in end-repairing reactions of T4 polynucleotide kinase (T4PNK), which is often included to phosphorylate DNA 5' terminus, but also has DNA 3' phosphatase activity[21]. Since the phosphatase activity of T4PNK requires magnesium and has an optimal pH of 6.0, which is different from its kinase activity (around pH 8.0)[21], care should be taken to utilize T4PNK for 3' phosphate removal.

The varying susceptibility of abnormal DNA ends to different enzymes provides a novel insight into the efficient 3' end-repair of DNA. Currently, commercially available kits for DNA end-repair may contain T4DP and T4 polynucleotide kinase (PNK). We tested a kit from New England Biolabs (EndRepair kit) and found that it worked to a limited extent, as far as the SAP-T4DP treatment could achieve, indicating that a fraction of the DNAs with 3' abnormal end remain unrepaired (see Supplementary Fig. 4 for EndRepair kit, and Supplementary Figs. 5 and 6 for T4PNK mediated FAM phosphorylation, needed to interpret the results shown in Supplementary Fig. 4). The lack of a complete reaction results from the fact that several abnormal ends, possibly 3'-recessed or blunt ends, are resistant to T4DP in the presence of dNTPs, which need to be added to prevent T4DP from degrading entire substrates. A solution here was to include an enzyme that has 3'–5' exonuclease activity. We named the 3'-end repair method via the combined activity of DNA polymerase and 3'–5' exonuclease as "SB repairing" (for scrap-and-build repairing).

A fraction of sonication-generated DNA termini was resistant, even after the removal of 3' phosphate, to T4DP in the presence of dNTPs. Therefore, for completeness, a 3'–5' exonuclease that can remove T4DP-resistant ends should be included. With regards to the possibility that the ratio of T4DP-resistant termini may differ among different samples, SB repairing could be an ideal method. It is also worth noting that exonuclease III can remove the 3' phosphoglycolate, generated by X-ray exposure[19]. The efficiency of SB repairing would be beneficial for analysis of samples, for which scarce amount of DNA could be used, for examples, DNA from a single cell and DNA recovered from fossils. It would also allow comprehensive analysis of DNA contained in a tube.

SB repairing method requires further improvements. In our experiments, we had to conduct two separate reactions. In the first step, the samples were simultaneously treated with T4DP and exonuclease III, followed by treatment with T4DP alone. Between these two steps, we conducted a DNA purification step to remove exonuclease III. The exclusion of the purification step will be conducted in future studies. However, optimization of the ratio of exonuclease III and T4DP would not be fruitful because dilution of exonuclease III would lead to insufficient repair, and increasing exonuclease III would lead to more recessed end. Our attempt to selectively inactivate exonuclease III by heating was not successful. Furthermore, addition of extra T4DP to the first reaction resulted in peak profiles that indicated the presence of 3'-recessed termini. The application of an antibody or a DNA aptamer that can bind and inhibit exonuclease III activity would be a potential solution.

In the repair reactions, enzymes that react with the 5'-end of DNA were not used, and the 5' termini generated by mechanical shearing remained unchanged. It is possible that 5'-terminal structures other than 5' phosphate or the 5'-OH terminus are generated, which will not be substrates for phosphorylation by T4PNK or DNA ligase. It should be noted that, although SB repairing is sufficient for G-tailing and CIS reactions to occur, for ligase-mediated adapter DNA ligation, repair and subsequent phosphorylation of 5' termini are required.

Provided that of 5'-OH plus 5' phosphate termini 95% carry the 5' phosphate group[12], and assuming that one strand breakage generates one phosphate terminus at most, that 46.7% of the 3' termini carried phosphate implies that about half of the 5' termini carry neither a -OH nor a phosphate group.

We used a capillary sequencer and the data obtained were analyzed using TraceViewer software. The resolution of the sequencer was as good as less than one nucleotide, and DNA molecules with different end structures could be discriminated by a slight change in the location of the peak top. We utilized this resolution to demonstrate that SAP-treated DNAs still carried at least two types of ends.

Currently, polyacrylamide gel electrophoresis under denaturing conditions is widely used in different studies to analyze DNA; however, most of those analyses could be performed by capillary sequencers, which can facilitate DNA-related studies. We have previously used this combination, i.e., a capillary sequencer and TraceViewer, to reveal and maximize the extraordinary high tailing activity of MMLV-RT[14,15], to find optimal conditions for the CIS reaction[16], to identify the nick site of a conjugative

plasmid NAH7[22], and to identify the DNA binding site of a transcriptional regulator by DNase I footprinting analysis[23].

## Methods

**Preparation of DNA with mechanically sheared ends**. A 560-bp DNA fragment was amplified by PCR using a primer pair, SA560 (6FAM-5'-AATGATACGGC GACCACCGAGATCTACAC-3') and SA645 (5'-CTGTCTCAGCATTTATCAG GGTTATTG-3') with plasmid DNA (pGiTp;[14]) as a template. Subsequently, the amplified product was treated with exonuclease I (TAKARA BIO Inc., Shiga, Japan) for 30 min at 37 °C to eliminate single-stranded DNAs, extracted with phenol/chloroform/isoamyl alcohol, and precipitated by ethanol precipitation. Subsequently, the DNA fragment was separated by polyacrylamide gel electrophoresis (PAGE), and a fluorescent DNA band was excised. DNA in the gel slice was recovered in Tris-EDTA (TE) buffer using a modified version of the crash-and-soak method[18]. In the method, the gel slice was extensively crashed in TE buffer for about 3 min, and after centrifugation the DNAs in the supernatant were purified by using DNA purification column.

To prepare the H4 fraction, the FAM-labeled DNA fragment was treated with model S220 acoustic solubilizer (Covaris Inc.) with the following settings: 10% duty-factor, 5 intensity value, 200 cycles/burst, and 60 s treatment time. The DNA was concentrated by ethanol precipitation and separated by PAGE. Then, various DNA size fractions, each containing a size-range width of approximately 50 nucleotides, were recovered by the crash-and-soak method. The concentration of the FAM-labeled DNA fragment mixture was measured fluorophotometrically by using an Infinite 200 fluorescence spectrophotometer (TECAN, Swiss). We used SA560 as a standard. The 560-bp DNA was also subjected for fragmentation with a milder set of sonication conditions than described above; 2% duty-factor, 4 intensity value, 200 cycles/burst, and 15 s treatment time, which are recommended by the manufacturer to fragment genomic DNAs into 1500 bp in average.

**Fragment analysis**. Two microliters of GeneScan™ 500 LIZ® size standard was added to 1 mL of HiDi-formamide (Thermo Fisher Scientific, US-MA) to prepare HiDiLIZ500. To 12.5 μL of the HiDiLIZ500, c.a. 1–25 fmol of DNA was added. The samples were analyzed using a model 3130xl sequencer with a 50 cm capillary and a POP7 polymer. The electro-injection time duration was extended from the default setting of 15 s to 90 s. The data file in ABIF format was analyzed by TraceViewer (http://www.ige.tohoku.ac.jp/joho/gmProject/gmdownload.html).

**Enzymatic reactions**. T4DP was purchased from TAKARA (Japan), and the provided reaction buffer was used (final concentration; 33 mM Tris-acetate pH 7.9, 66 mM $CH_3COOK$, 10 mM $(CH_3COO)_2Mg$, 0.5 mM dithiothreitol). Unless stated otherwise, dNTPs were added at a final concentration of 0.25 mM each, and 0.5–2.5 units of T4DP were used in 10 μL-reactions at 37 °C for 5 min. Exonuclease III was purchased from TAKARA (Japan) and diluted 10-fold in 25 mM Tris-HCl (pH 8.0), 50 mM KCl, 0.5 mM dithiothreitol, and 50% glycerol. Exonuclease III was used in conjunction with T4DP in the T4DP buffer conditions. The reaction was conducted at 37 °C for 30 min.

Shrimp alkaline phosphatase (SAP) was purchased from Roche. SAP reaction was conducted at 37 °C for 30 min. TdT was purchased from TAKARA (Japan), and the provided reaction buffer was used with BSA at a final concentration of 0.01%. Reverse transcriptase from Molony Mouse Leukemia Virus (MMLV-RT) was purchased from Nippon Gene Co. Ltd. (Tokyo, Japan). Buffer compositions are as described previously[16]. The G-tailing reaction mixture contained, in a total volume of 10 μL, 50 fmols of substrate DNA, 50 mM Tris-HCl pH 8.3, 75 mM KCl, 2 mM DTT, 6 mM $MgCl_2$, 4 mM dGTP, 4 mM $MnCl_2$ and 50 U MMLV-RT. The CIS reaction mixture contained substrate DNA, 50 mM Tris-HCl pH 8.3, 75 mM KCl, 2 mM DTT, 6 mM $MgCl_2$, 0.25 mM each of dNTPs, a singlestranded DNA, and 50 U MMLV-RT. For reproducibility, we took special care on the freshness of DTT, as well as on the order of the addition of component, i.e., we added enzyme as the last component, and $MgCl_2$ was added just before the enzyme addition. This is because addition of $MgCl_2$ to unbuffered acidic conditions formed white insoluble material.

The enzymatic reaction products were purified by DNA binding column purification and analyzed using a capillary sequencer, except for T4DP-treated samples in the absence of dNTPs, for which degradation products were to be observed.

**G-tailing and CIS reactions**. G-tailing and CIS reactions were conducted as previously described[15,16], except that G-tailing was conducted at 37 °C for 1 h. For each CIS reaction 2 pmol of a single-strand DNA SA718 (biotin-5'-GTGACTGG AGTTCAGACGTGTGCTCTTCCGATCTNNNNNNNNNDDCCCCCCC-3') was used as a guide adapter oligonucleotide. To initiate CIS reaction, to a mixture of G-tailed DNA and SA718 placed at room temperature (25 °C), enzyme mix containing all of the other components were added and mixed, and then transferred to 37 °C and incubated. Here, 5' biotin was used to prevent concatemer formation[16], which may otherwise compromise the results.

**DNA purification**. To purify DNA, the HiYield Gel/PCR DNA Fragments Extraction Kit (RBC Bioscience) was purchased and mini-columns were made by breaking DNA binding columns and refilling the column matrix into filtered tips (VistaRak cat No:4060-1333 VistaLab Technologies Inc.) whose tip was trimmed just beneath the filter. Eight to twelve min columns were made from a single column. To fit the mini column to a 2-mL collection tube, two kinds of adapter were made. One was made by cutting 0.5 mL tubes into top and bottom parts, and the top part was used. The other was made by cutting the tip rack. We eluted DNA to 0.5 mL tube with 2 to 5 μL of water or TE buffer. For reproducibility, we avoided too long incubation of DNA with salt solution (DF buffer) before column passage (each sample was mixed with the salt solution and immediately placed on a column and centrifuged). The DNA samples purified after prolonged incubation seem to contain co-purified dNTPs that prevented DNA from degradation upon T4 DNA treatment that was designed to be conducted in the absence of dNTPs. To obtain stable TdT reaction results, the DNA binding column was vacuum-dried before elution, and less than 1/5 of the elution fraction was used.

**Reporting summary**. Further information on research design is available in the Nature Research Reporting Summary linked to this article.

## Data availability

All relevant electropherogram data are provided upon request by Y.O. Those electropherogram data in ABIF format are stored in our laboratory.

## Code availability

TraceViewer software is available at http://www.ige.tohoku.ac.jp/joho/gmProject/gmdownload.html

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

## Acknowledgements

This work was supported by Grants-in-Aid from the Ministry of Education, Culture, Sports, Science, and Technology, Japan (15H04471) and by a grant from the Institute for Fermentation, Osaka (IFO), Japan.

## Author contributions

Y.O. conceived and designed the project, performed analyses, wrote the code for TraceViewer, and wrote the paper. All authors, Y.O., K.S., Y.N. and M.T. contributed to discussions, editing, and revision of the paper.

## Competing interests

The authors declare no competing non-financial interests but the following competing financial interests; Tohoku University filed domestic patent applications on the tailing and CIS reactions.
