## [Peer Review File · Communications Biology]

Reviewers' comments:

Reviewer #1 (Remarks to the Author):

Sharing DNAs by sonication is one of the most common steps of NGS library preparation. However, the structure of 3' ends of the DNA fragments remains unclear, and the way to heal 3' ends is a subject of improvement. Ohtsubo et al. found that 3'-OH ends occupied only 20% of total DNA that is shared by sonication and the others include 3' phosphate (45%) and abnormal termini of unidentified (35%). They invented efficient ways to repair mechanically shared 3' DNA ends, which utilize Shrimp alkaline phosphatase (SPA) to remove 3' phosphate or Exonuclease III (ExoIII) to degrade 3' region of DNA. They refer the method using ExoIII and T4 DNA polymerase as scrap-and-build repairing (SB-repairing).

Major points:

1. It would be better to summarize the properties of all the enzymes they used in this manuscript, in the introduction or the results.
2. It would be better to explain under which conditions the efficient repair of mechanically shared 3' DNA ends becomes very important. e.g. single cell analysis and rare DNA samples of ancient animals and plants.
3. The formation of 3' phosphate and abnormal termini is biased in the context of DNA sequence? If it is so, it is necessary to repair damaged 3' ends to obtain unbiased results in DNA sequencing and ChIP analysis.
4. The following sentence is not easy to follow in the abstract. It is better to rewrite it. "Notably, over 50% of the abnormal termini (about 20% of the total) were not repaired after removal of 3' phosphates and T4 DNA polymerase (T4DP) treatment." (lines 36-38)"
5. Can the authors provide a conclusion from the following observation: "The electropherogram exhibited a disordered pattern, as seen by...." (lines 172-173)
6. ", suggesting that the 3' termini changed towards a normal (3'-OH) state." (lines 194-195). It would be better to clarify that this interpretation came from the comparison between Fig. 1C and E.
7. Can the authors provide more detailed explanation of the following sentence: "However, since this change seems likely...., which are not suitable substrates for the downstream reactions with different enzymes" (lines 195-198). Why the 3' abnormal structure in blunt or recessed configuration? In what sense they are not suitable substrates? What would be the downstream reactions with different enzymes?
8. "SAP treatment for 30 min resulted in disorderly spaced peaks (Fig. 1D)." (line 186). This sentence needs to be changed because the authors wrote in the earlier part that "the electropherogram exhibited a disordered pattern" (line172). It is unclear whether the authors try to argue that the SAP treatment changed the electropherogram pattern or not.
9. Because the standard way to treat the DNA ends after sonication during the preparation of NGS library using T4 DNA polymerase, it is better to include the result of T4 DNA polymerase-TdT (T4DP-TdT) in Figure 3.
10. Would the authors explain the difference between SAP-TdT and SAP-T4-TdT results in Figure 3.
11. Would the authors explain the reason why it is important to evaluate the repaired DNA termini by two different ways (i.e. TdT and G-tailing/CIS)
12. It is better to perform some kind of quantitative analysis to compare the E3T4-T4DP and SAP-T4DP results (Fig. 4C and D).
13. Fig. 3 and 4 show that SAP treatment is not enough to repair 3' ends completely and that the treatment of DNA with ExoIII is more effective. However, Fig 5 shows that SAP treatment is enough to make almost all DNAs sensitive to the exonuclease activity of T4 DNA polymerase. Would the authors explain what causes this difference.

Minor points:

1. It would be better to avoid uncommon abbreviations: T4DP, MSDEs, and MSD3Es.
2. To make it clear, please spell out "NGS" (line 56), "CIS" (lines 82), "PAGE" (line 112), "TE" (line 113), "DTT" (line 135) when they appear in the text for the first time.
3. "Elsner et al." (lines 64, 73, and 289) should be "Elsner and Lindblad".
4. Cite a reference paper for the crash-and-soak method (lines 113-114).
5. Name the equipment that is used to measure the DNA concentration (lines 120-121).
6. Can you explain the detailed background of "less than 1/5 of the elution fraction was used" (line 158).
7. There are strange characters in the legend of Figure 2.
8. The authors should provide the data as supplementary information (lines 183 -185).
9. The authors need to refer Fig. 4A, 4B, and Fig. 5A somewhere in the text.
10. In Figure 5, can the authors explain why the 5'-FAM-AA-3' (***) ran earlier than 5'-FAM-A-3' (*).
11. I recommend to cite the following paper that describe the standard method to repair DNA ends during the preparation of Illumina sequencing library. Meyer and Kircher 2010 (PMID: 20516186).

Reviewer #2 (Remarks to the Author):

In this paper, Ohtsubo et al systemically studied the potential DNA terminus (3') generated by sonication. By using different biochemical assay, the authors found that around 20% of DNA terminus generated by sonication had neither 3' phosphate group nor 3' hydroxyl group. End repairing of all different kinds of DNA terminus were then achieved by the novel method using a combination of exonuclease III and T4 DNA polymerase, followed by another round T4 DNA polymerase filling in reaction.

Overall, the topic of this manuscript is very interesting and important. The novel method provides a useful strategy for DNA handling that includes sonication. Several points need to be addressed before publication:

1. In this paper, it is found that 20% sonicated DNA had abnormal ends. Do the authors think that this percentage is DNA length and/or sonication intensity dependent? In the references with different conclusion (#9 and #11), it seems that the original DNA length is much longer and the sonicated DNA products are still a couple of kilobases. Moreover, it will be better if the authors can explain the reason to choose H4 fraction. Does other fractions have similar 3' terminals with H4?
2. From Fig 1, can the author find that abnormal ends happen in a sequence-specific manner? If so, the current end-repairing method (such as SAP-T4DP) will have a selective power on DNA fragments and provide biased results, which will also make the novel end-repairing method (E3T4DP-T4DP) more important. Moreover, the authors should include their data with the commercial EndRepair Kit from NEB. This kit can be served as a great control to show the significance of the novel method (E3T4DP-T4DP) in all of the assays.
3. In page 11, the authors mention that they tested different ratios of E3 and T4DP (line 211 to 213). It seems the current combination isn't perfect and requires another round of T4DP treatment. Therefore, including the data for other combinations will be helpful for further optimization.
4. In Fig 3 and 4, these two assays serve as the same function to evaluate end-repairing efficiency. I am wondering if the authors have specific reason to organize these two experiments in a different way. For example, Fig 3 has no T4DP treated samples and Fig 4 has no SAP treated samples. Moreover, the order of each panel and labeling (T4 in Fig 3 and T4DP in Fig 4) are also inconsistent.

5. In Fig. 5, the two peaks around 50 nt and 75 nt are considered as 2 nt and 1 nt, respectively. Can the authors explain why the bigger one, 2 nt, is close to 50 nt and the smaller one, 1 nt is close to 75 nt? Moreover, Fig 2D with Exo3 treatment shows different peaks, which are smaller than 35 nt; what could these peaks be?

6. In page 17, the authors suggest "T4DP can degrade ... in a sequence-specific manner" (line 261 to 265). Can the authors label the sequence for Fig. 5C to show the specific sequence? Moreover, besides diminished peaks in Fig. 5C, there are also some peaks with increased height, can the authors explain what these could be?

First of all, we would like to sincerely thank reviewers for constructive comments and suggestions. Followings are our item by item responses to the points raised. The changes made are highlighted in red in the revised manuscript.

>Reviewer #1 (Remarks to the Author):

>Sharing DNAs by sonication is one of the most common steps of NGS

>library preparation. However, the structure of 3' ends of the DNA fragments

>remains unclear, and the way to heal 3' ends is a subject of improvement.

>Ohtsubo et al. found that 3'-OH ends occupied only 20% of total DNA that

>is shared by sonication and the others include 3' phosphate (45%) and

>abnormal termini of unidentified (35%). They invented efficient ways to

>repair mechanically shared 3' DNA ends, which utilize Shrimp alkaline

>phosphatase (SPA) to remove 3' phosphate or Exonuclease III (ExoIII) to

>degrade 3' region of DNA. They refer the method using ExoIII and T4 DNA

>polymerase as scrap-and-build repairing (SB-repairing).

>Major points:

>1. It would be better to summarize the properties of all the enzymes they

>used in this manuscript, in the introduction or the results.

We agree that use of different kinds of enzymes makes the manuscript hard to follow, and summarizing the properties would be helpful. However, even if we summarize once, we should describe relevant characteristics again to interpret each experimental result, and this would make the manuscript redundant. We believe that relevant characteristics of each enzyme are stated in the text as needed.

>2. It would be better to explain under which conditions the efficient repair of

>mechanically shared 3' DNA ends becomes very important. e.g. single cell

>analysis and rare DNA samples of ancient animals and plants.

We are grateful for the suggestion. We added a couple of sentences in a section of the discussion (lines 372 to 376 in the revised manuscript).

>3. The formation of 3' phosphate and abnormal termini is biased in the

>context of DNA sequence? If it is so, it is necessary to repair damaged 3'

>ends to obtain unbiased results in DNA sequencing and ChIP analysis.

We are thankful for the constructive question. So far, we have not observed the DNA sequence dependency. At least there seems to be no strong dependency. (Unreacted peaks in Fig. 4C and Fig. 5B are abnormal and 3' phosphate termini, respectively, and their distributions seems not to be biased). We added a

sentence in a section of the discussion (lines 306 to 308 in the revised manuscript).

>4. The following sentence is not easy to follow in the abstract. It is better to
>rewrite it. "Notably, over 50% of the abnormal termini (about 20% of the
>total) were not repaired after removal of 3' phosphates and T4 DNA
>polymerase (T4DP) treatment." (lines 36-38)

We believe the repeated use of "%" caused the problem. We replaced "over 50%" with "more than half".

>5. Can the authors provide a conclusion from the following observation:
> "The electropherogram exhibited a disordered pattern, as seen by...."
> (lines 172-173)

We added a sentence as suggested (lines 184 to 185).

>6. ", suggesting that the 3' termini changed towards a normal (3'-OH)
>state." (lines 194-195). It would be better to clarify that this interpretation came
>from the comparison between Fig. 1C and E.

We are thankful for the suggestion. We revised our manuscript accordingly. (line 208)

>7. Can the authors provide more detailed explanation of the following
>sentence: "However, since this change seems likely...., which are not
>suitable substrates for the downstream reactions with different enzymes"
> (lines 195-198). Why the 3' abnormal structure in blunt or recessed
>configuration? In what sense they are not suitable substrates? What would
>be the downstream reactions with different enzymes?

We are grateful for the point raised. As we have examined this part of the manuscript we found a gap between the sentence starting with "However, since..." and the sentence one before. We replaced the sentence. (lines 208 to 212 in the revised manuscript)

>8. "SAP treatment for 30 min resulted in disorderly spaced peaks
> (Fig. 1D)." (line 186). This sentence needs to be changed because the
>authors wrote in the earlier part that "the electropherogram exhibited a
>disordered pattern" (line172). It is unclear whether the authors try to argue
>that the SAP treatment changed the electropherogram pattern or not.

The SAP treatment resulted in altered but still disorderly spaced pattern. We inserted "altered but still" to indicate that the peak pattern has changed (line 199).

>9. Because the standard way to treat the DNA ends after sonication during
>the preparation of NGS library using T4 DNA polymerase, it is better to
>include the result of T4 DNA polymerase-TdT (T4DP-TdT) in Figure 3.
DNA EndRepair kit from NEB possibly contains T4 DNA polymerase and T4
polynucleotide kinase, which has 3' phosphatase activity. The inefficiency of T4
DNA polymerase treatment on sonication-generated DNA termini is shown by T4
DNA polymerase -G-tailing -CIS reaction in new Figure 3.

>10. Would the authors explain the difference between SAP-TdT and
>SAP-T4-TdT results in Figure 3.

We added a couple of sentences to explain the difference (lines 258-262). We
are grateful for the suggestion for improvement. In the revised manuscript the
figure is now figure 4.

>11. Would the authors explain the reason why it is important to evaluate the
>repaired DNA termini by two different ways (i.e. TdT and G-tailing/CIS)

This point was also suggested by Reviewer 2. The use of TdT is to complement
the limitation of the quantitative analysis by G-tailing/CIS (see also our response
to the next point raised). In our course of experiments, the experiment shown in
old Fig.4 was conducted earlier than that shown in old Fig.3, and the latter was
designed to compensate the quantitative limitations. We exchanged the Figures
3 and 4, and explained the quantitative limitations. Several parts were revised
and changes were highlighted in red.

>12. It is better to perform some kind of quantitative analysis to compare the
>E3T4-T4DP and SAP-T4DP results (Fig. 4C and D).

Unfortunately, to soundly conduct quantitative analysis of the results shown in
new Fig.3C and 3D was impossible. This is because of the nature of the H4
fraction, which contained smaller DNAs (see the signals around 75 nt of new Fig.
3B). The CIS products deriving from the smaller DNAs are located around 100 nt
- 150nt. This is in contrast to the experiment using TdT, which extends the 3' OH
termini extensively. This is the reason we also used TdT. We added a sentence
(lines 246-248) to describe this.

>13. Fig. 3 and 4 show that SAP treatment is not enough to repair 3' ends
>completely and that the treatment of DNA with ExoIII is more effective.
>However, Fig 5 shows that SAP treatment is enough to make almost all
>DNAs sensitive to the exonuclease activity of T4 DNA polymerase. Would

>the authors explain what causes this difference.

In the original manuscript we described " the H4 DNA was reacted with T4DP in the absence of dNTPs, a condition that promotes 3'->5' exonuclease activity of T4DP". To make it clearer, we added a phrase (lines 280-281) to explain that abnormal 3' termini are sensitive to the exonuclease activity of T4DP in the absence of dNTPs. Note that applying dNTP in the absence of dNTP removes abnormal termini but at the same time degrades entire DNA in a hardly controllable manner.

>Minor points:

>1. It would be better to avoid uncommon abbreviations: T4DP, MSDEs, and >MSD3Es.

Those abbreviations might not be common, however we believe that those abbreviations would make each sentence easier to understand. No changes have been made.

>2. To make it clear, please spell out "NGS" (line 56), "CIS" (lines 82), > "PAGE" (line 112), "TE" (line 113), "DTT" (line 135) when they appear in the >text for the first time.

We followed the suggestion.

>3. "Elsner et al." (lines 64, 73, and 289) should be "Elsner and Lindblad".

We followed the suggestion.

>4. Cite a reference paper for the crash-and-soak method (lines 113-114).

We cited a reference, and to be more precise, we added a sentence to depict our modification (lines 116-118).

>5. Name the equipment that is used to measure the DNA concentration > (lines 120-121).

We revised our manuscript (lines 125-126).

>6. Can you explain the detailed background of "less than 1/5 of the elution >fraction was used" (line 158).

We experienced failure to observe TdT-extended products several times, and came to a conclusion that the failure is due to carry-over of wash buffer used in DNA column purification. Although we did not establish the adverse effects of the wash buffer, for the reproducibility of our result, we included this information. The text remains unchanged.

>7. There are strange characters in the legend of Figure 2.

Those characters are "μ". We corrected.

>8. The authors should provide the data as supplementary information (lines 183 -185).

We added a supplementary Fig. S1. (The previous Fig. S1 is now Fig. S2)

>9. The authors need to refer Fig. 4A, 4B, and Fig. 5A somewhere in the text.

We referred to those as suggested.

>10. In Figure 5, can the authors explain why the 5'-FAM-AA-3' (**) ran earlier than 5'-FAM-A-3' (*).

We added a sentence to the legend of Fig. 5.

>11. I recommend to cite the following paper that describe the standard method to repair DNA ends during the preparation of Illumina sequencing library. Meyer and Kircher 2010 (PMID: 20516186).

We cited the reference as recommended.

Reviewer #2 (Remarks to the Author):

>In this paper, Ohtsubo et al systemically studied the potential DNA terminus (3') generated by sonication. By using different biochemical assay, the authors found that around 20% of DNA terminus generated by sonication had neither 3' phosphate group nor 3' hydroxyl group. End repairing of all different kinds of DNA terminus were then achieved by the novel method using a combination of exonuclease III and T4 DNA polymerase, followed by another round T4 DNA polymerase filling in reaction.

>Overall, the topic of this manuscript is very interesting and important. The novel method provides a useful strategy for DNA handling that includes sonication. Several points need to be addressed before publication:

>1. In this paper, it is found that 20% sonicated DNA had abnormal ends. Do the authors think that this percentage is DNA length and/or sonication intensity dependent? In the references with different conclusion (#9 and #11), it seems that the original DNA length is much longer and the sonicated DNA products are still a couple of kilobases.

We are thankful for the suggestion. We conducted an additional experiment, in which the FAM-labeled 560 bp DNA was treated by covaris under a mild setting recommended by the manufacturer to fragment genomic DNAs into 1500 bp in average. After the treatment, we observed that most of the 560 bp fragment remained unchanged but a fraction of the DNA was fragmented resulting in peaks of sizes about 100 nucleotides to 400 nucleotides. As expected the peaks were disorderly spaced, and the ends were recalcitrant to end-repairing by T4DP, and E3T4-T4DP treatment resulted in the orderly spaced peaks. We added a sentence to describe this to a section of discussion (lines 308-311).

>Moreover, it will be better if the authors can explain the reason to choose
>H4 fraction. Does other fractions have similar 3' terminals with H4?
We analyzed different fractions to reach the findings and to establish experimental conditions, and then used the H4 fraction for full analysis. We added a sentence to describe that we obtained consistent data using different fractions (lines 176-178).

>2. From Fig 1, can the author find that abnormal ends happen in a
>sequence-specific manner? If so, the current end-repairing method (such
>as SAP-T4DP) will have a selective power on DNA fragments and provide
>biased results, which will also make the novel end-repairing method
> (E3T4DP-T4DP) more important.

We are grateful for the constructive suggestion. In the new Figure 4C, the remaining peaks represent DNAs with abnormal end. The profile of the remaining peaks indicates that generation of the abnormal ends seems not to be sequence specific.

>Moreover, the authors should include their data with the commercial EndRepair
>Kit from NEB. This kit can be served as a great control to show
>the significance of the novel method (E3T4DP-T4DP) in all of the assays.
As we have described in the original manuscript, the NEB EndRepair kit worked to a limited extent as the SAP-T4DP treatment. To show this, we added a supplementary Fig. S3. The assessment of EndRepair kit was not straight forward, because T4PNK possibly contained in the kit had a weak activity to phosphorylate FAM, and affected the electro-mobility of the strand. The incomplete reaction brought about by the weak activity, created a mixture of FAM-phosphorylated and FAM-non-phosphorylated strands, resulting in a disordered pattern. We explained it in the legend to Fig. S3.

>3. In page 11, the authors mention that they tested different ratios of E3 > and T4DP (line 211 to 213). It seems the current combination isn't perfect >and requires another round of T4DP treatment. Therefore, including the >data for other combinations will be helpful for further optimization. We experienced that diluting E3 too much resulted in imperfect polishing. Possibly abnormal ends are bad substrate of E3. On the other hand using too much E3 resulted in recessed end. Most likely, optimization would not be achieved by modifying the ratio. To emphasize that ratio-based optimization would not be fruitful, we modified the discussion (lines 382- 385).

>4. In Fig 3 and 4, these two assays serve as the same function to evaluate >end-repairing efficiency. I am wondering if the authors have specific reason >to organize these two experiments in a different way.

Reviewer 1 also raised a similar point. See our response to the point #11 of reviewer 1

>For example, Fig 3 has no T4DP treated samples and Fig 4 has no SAP >treated samples. Moreover, the order of each panel and labeling (T4 in Fig >3 and T4DP in Fig 4) are also inconsistent.

We believe that the problem here was that the two experiments seemed to be conducted in parallel but their experimental details are not parallel, and might be confusing. As we have revised the manuscript, now these two experiments are not parallel, and now it is less confusing. We corrected the labeling inconsistencies all across the manuscript (E3T4-T4 was replaced by E3T4-T4DP, and SAP-T4 was replaced by SAP-T4DP).

>5. In Fig. 5, the two peaks around 50 nt and 75 nt are considered as 2 nt >and 1 nt, respectively. Can the authors explain why the bigger one, 2 nt, is >close to 50 nt and the smaller one, 1 nt is close to 75 nt?

We added a sentence to the legend of Fig. 5.

>Moreover, Fig 2D with Exo3 treatment shows different peaks, which are >smaller than 35 nt; what could these peaks be?

Those peaks represent small fragments with 5' FAM. The lack of signals around 50 nt and 75 nt might be due to different enzymatic properties of exonuclease III, whose final product might be larger than FAM + two-nucleotides.

>6. In page 17, the authors suggest "T4DP can degrade ... in a >sequence-specific manner" (line 261 to 265). Can the authors label the

>sequence for Fig. 5C to show the specific sequence?

Here, we intended to explain why we used the data obtained after 5 minutes for quantification and not meant to insist sequence specific degradation by T4DP. We supposed that the activity of T4DP to degrade DNA with 3' phosphate was over-discussed in the previous manuscript, and revised the manuscript accordingly. We are unable to label the sequence in Fig.5C. This is because 3' phosphate affects the electro-mobility, and we have no technic to create a size standard with 5'-FAM and 3'-phosphate.

>Moreover, besides diminished peaks in Fig. 5C, there are also some peaks

>with increased height, can the authors explain what these could be?

We are grateful for the reviewer 2 for raising an important point. Theoretically, amount of each 3' phosphate DNA fragment should never increase. We examined the original data. Each sample was analyzed four times, and two peaks adjacent to each other located in the middle of 100 nt and 138 markers are (apparently) getting higher in all of the four replicates. We noticed that signals from the 201 bp FAM label DNA are diminished for the 20 minutes samples, raising a possibility that y-axis scaling using the 201 bp DNA is not perfect. Most likely, the injection profile has changed by the accumulation of the small FAM-nucleotides or by slight buffer condition change brought about by the DNA strand degradation. Since the quantitative values are obtained where the putative injection profile change is not marked, they should not be affected by this effect. We added a sentence to the legend of Fig .5 to describe the apparent hight increase.